# Epidemiological, clinical and laboratory profile of patients presenting with severe acute respiratory syndrome (SARS-CoV-2) in Ethiopia

**Addisu Gize** [1,2]*, **Melkayehu Kassa**[1], **Solomon Ali**[1], **Yosef Tadesse**[3], **Bereket Fantahun** [4], **Yitagesu Habtu**[5], **Aman Yesuf**[6]

1 Department of Microbiology, School of Medicine, St. Paul's Hospital Millennium Medical College, Addis Ababa, Ethiopia, 2 CIH[LMU] Center for International Health, LMU University Hospital, LMU Munich, Germany, 3 Department of Anatomy, School of Medicine, St. Paul's Hospital Millennium Medical College, Addis Ababa, Ethiopia, 4 Department of Pediatrics, School of Medicine, St. Paul's Hospital Millennium Medical College, Addis Ababa, Ethiopia, 5 School of Public Health, College of Health Sciences, Addis Ababa University, Addis Ababa, Ethiopia, 6 Department of Epidemiology, School of Public Health, St. Paul's Hospital Millennium Medical College, Addis Ababa, Ethiopia

* addisu.gize@sphmmc.edu.et

**Data Availability Statement:** All relevant data are within the manuscript and its Supporting information files.

## Abstract

### Introduction

Data regarding patients presenting with severe acute respiratory syndrome (SARS-CoV-2) illness have not adequately been documented which provides distinct insights into low-resource settings like Ethiopia. Thus, the study aimed to compare epidemiological, clinical and laboratory profiles of patients presenting with acute respiratory syndrome illness in Addis Ababa Ethiopia.

### Methods

We used a comparative cross-sectional study design among patients with SARS-CoV-2 illness at St. Paul's Hospital Millennium Medical College (SPHMMC), Addis Ababa, Ethiopia from October 2020 to September 2021. Using a structured questionnaire a consecutive sampling technique was applied to collect socio-demographic data. Additionally, nasal swabs were collected to confirm SARS-CoV-2 infection using a Real-Time Polymerase Chain Reaction. Blood samples were also collected from the participants for laboratory profiles (hematological tests like; white blood cell count, hematocrit, and platelet count; and biochemical and enzymatic tests like; aspartate transaminase (AST), creatinine, etc) analysis. Data were entered and analyzed using SPSS version 23.0 and p-values ≤0.05 were considered as statistically significant.

### Results

Of the total 413 participants presenting with SARS-CoV-2 illness, 250 (60.5%) participants tested positive for COVID-19 disease. COVID-19 patients were less likely to use an alcohol-

**Funding:** This research was funded by the St. Paul's Hospital Millennium Medical College. The funder had no role of the in designing of the study, collection, analysis, and interpretation of data.

**Competing interests:** The authors have declared that no competing interests exist.

based method of hand washing (12.5% vs 87.5%; p = 0.048). The COVID-19 patients had a higher proportion of headache (67.3% vs 32.7%, p = 0.001), sore throat (72.5% vs 27.5%, p = 0.001), and loss of sense of taste (74.4% vs 25.6%, p = 0.002). Patients with COVID-19 have significantly higher neutrophil than their counterparts (68.2% vs 31.8%; p = 0.001). Similarly, creatinine (64.9% vs 35.1%, p = 0.001) from renal function and alkaline phosphatase (66.8% vs 33.2%, p = 0.046) in the liver function tests were significantly higher in the COVID-19 patients.

## Conclusion

Our findings suggest the need to substantially consider headache, sore throat, and loss of taste as potential clinical diagnostic symptoms for early screening and testing. Elevation of neutrophil, creatinine, alkaline phosphatase profiles are also used for potential diagnostic biomarkers in screening and testing suspected patients.

## Introduction

Coronavirus disease 2019 (COVID-19) is an infectious disease caused by Severe Acute Respiratory Syndrome Corona Virus 2 (SARS-CoV-2), still a global concern [1, 2]; a novel ribonucleic acid (RNA) beta coronavirus [1, 3, 4]. In most cases, the transmission of COVID-19 occurs through respiratory droplets generated by infected patients during coughing, talking, and sneezing. It can also be transmitted indirectly through touching contaminated surfaces by hand with subsequent self-inoculation to the nose, mouth, or eye [5]. Despite the universal implementation of non-pharmacological COVID-19 preventive measures and the introduction of different preventive vaccines, the COVID-19 pandemic is still ongoing and a significant number of new infections and deaths are being reported every day. As of May 24, 2022, an estimate of more than five hundred twenty six million detected cases, more than six million deaths, and more than eleven billion administered vaccine doses were reported globally. As of May 2022, the African Centers for Disease Control and Prevention (CDC) has reported 11,580,083 detected cases and 252,829 deaths [6, 7].

Ethiopia has adopted a targeted testing strategy and detected cases with mild or moderate symptoms were quarantined in isolation centres and at home until the infection resolved. The COVID-19 treatment centre at Saint Paul's Hospital Millennium Medical College (SPHMMC) is one of the main public health facilities established solely to respond to the pandemic [8, 9]. The centre serves as an isolation and treatment centre and it has provided clinical, laboratory, and psychological care services for COVID-19 patients who came from various corners of the capital city of the country and outside of the capital city during the pandemic season.

The clinical spectrum of COVID-19 diseases ranges from asymptomatic or mild to severe life-threatening conditions. However, most of the clinical pictures presented by symptomatic patients are similar to other respiratory disease manifestations. In global literature, it is described that cough [2, 10–15] and dyspnea [16–18] and influenza-like symptoms like fever [2, 10–15] and myalgia [16–19] were the most common presenting symptoms. Elevation of laboratory profiles of COVID-19 patients including C-Reactive Protein (CRP) [13, 20], interleukin-6 [13] platelets, eosinophils, hemoglobin, and albumin [13, 20] also were described. Epidemiological data including contact history, familial clustering, co-morbidities [13], older ages [14], duration of symptoms and admission [19, 21, 22] were associated with COVID-19 positivity.

The majority of the global studies [10–12, 14, 19, 20] described the epidemiology, clinical, and laboratory profiles of COVID-19 symptomatic patients from high-income settings. However, the epidemiological, clinical, and laboratory profile data of COVID-19 patients managed at low resource settings might be quite different from what was described. We hypothesized that the epidemiology, relative proportion of clinical profiles and laboratory profiles of COVID-19 patients are different for different settings associated with socio-demographic, quality of medical care, nutrition, and genetic and immunological factors. In addition, as to our knowledge analogous data have not been previously published in a peer-reviewed journal and we believe the conclusions provide distinct insights that are of relevance to a similar context. Thus, locally generated evidence on the epidemiology, clinical profile and laboratory profile of COVID-19 patients is a key to early suspicion and identification of COVID-19 patients, prioritizing resources and tailoring the management. Furthermore, the continued emergence of new variants of SARS-CoV-2 is also associated with a change in the clinical presentations of the disease, which requires monitoring of the clinical profile of patients.

Thus, this study aims to generate evidence about the most pertinent epidemiologic features, clinical profiles and laboratory findings COVID-19 patients managed in low-resource settings like in Ethiopia to guide the testing and treatment strategies.

## Methods and materials

### Study setting

The study was conducted on one of main COVID-19 center of the country, SPHMMC in Addis Ababa, Ethiopia during the SARS-CoV-2 pandemic time for those presenting respiratory illness. The hospital is a referral specialized hospital in Addis Ababa. Patients presenting with SARS-CoV-2 came to the centre from various corners of the country outside of the capital city of Ethiopia. The hospital is expected to serve estimated total population of more than 5 million peoples. We included all patients 18 years above who underwent testing for SARS-CoV-2 within 24 hours of presentation to this referral hospital. Suspected patients were tested for SARS-CoV-2 if they met COVID-19 symptoms clinically or laboratory testing criteria of the National Ethiopian Public Health Guideline [23].

### Study design

One year, from October 2020 to September 2021 comparative cross-sectional study was conducted for all COVID-19 suspected patients.

### Study participants

The study included any acute respiratory illness (runny nose and sore throat) and at least one of the following symptoms: fever, cough, and shortness of breath. All suspected COVID-19 patients who have essential clinical features of acute respiratory illnesses or the most common presenting symptoms of acute respiratory illnesses (cough and dyspnea), influenza-like symptoms (fever and myalgia) and known contact history with COVID-19 patients were included in the study, whereas previously known COVID-19 disease patients were excluded from the study. The first encounter of the patient was considered for inclusion in the study if the patient had multiple encounters within the study period. Multiple clinical encounters were considered if the patient was discharged and readmitted after three days in the study period.

## Sample size and sampling procedure

The sample size used for this study was based on the proportion of patients with COVID-19 positivity status. Because we had no previous proportion of positivity during the study period, we took the positivity rate of RT-PCR COVID-19 to be 50%, with a 5% margin of error, a 95% confidence level, and a 10% non-response rate and keeping the assumption of a single population proportion formula.

$$n = \frac{(z^{\alpha}/_2)^2 p(1-p)}{d2} \Rightarrow \frac{(1.96)^2 0.5(1-0.5)}{(0.05)2} = 384 \text{ study subjects.}$$

Where: n = minimum sample size,
P = estimated proportion of patients with SARS-CoV-2 in the study population, and taking 10% non-response rate, the final sample size become 423 participants.
d = the margin of sample error, $z^{\alpha}/_2$ = the standard normal variable at $1-^{\alpha}/_2$ confidence level and we used consecutive sampling technique was used to select the study population.

Therefore, the final sample size was determined to be 423 study subjects. We used a consecutive sampling technique suspected individuals per day from the isolation centre of SPHMMC during the study period.

# Data collection and laboratory procedures

## Socio-demographic and clinical data

A pre-tested questionnaire and check-list were used to collect socio-demographic status and clinical and laboratory tests of the study participants. The data was collected by trained physicians and laboratory technologists, and all authors had no access to information that could identify individual participants during or after data collection. Accordingly, patient clinical data on initial clinical presentation, comorbidities, and relevant treatment and clinical outcomes for all patients presented with acute respiratory symptoms or influenza-like illness symptoms were recorded using the prepared formats for each department. Additional information on patient demographics, vital signs, and laboratory results were obtained from the medical records.

## COVID-19 testing method

After the study subjects met the COVID-19 clinical and microbiological laboratory testing criteria of the Ethiopian Public Health Institute Guideline (EPHI) [23], oropharyngeal (OP) and/ or nasopharyngeal (NP) swabs were collected to confirm the disease using RT-PCR assay at the SPHMMC testing centre.

Then NP and OP specimens were mixed in a single tube to maximize test sensitivity. Then, the mixed sample was transported to the COVID laboratory in VTM (in cold chain 2–8˚C) for RT-PCR analysis of SARS-CoV-2.

## RNA extraction and RT-PCR analysis

Two hundred microliter (200μL) of the combined swab (NP & OP) was mixed with 50 μl proteinase K and 200μl lysis buffer that contains a guanidinium-based inactivating agent and then viral RNA was extracted using a nucleic acid isolation Kit (Da'an Gene Corporation), China. Then, viral RNA was eluted with 60μL elution buffer and RT-PCR reagent (Da'an Gene Corporation) was employed for SARS-CoV-2 detection. Also, two PCR primer and probe sets, which target the open reading frame 1a/b (ORF 1a/b) (FAM reporter) and nucleocapsid protein genes and VIC reporter genes were added to the same reaction mixture. In each run,

positive and negative controls were included. Samples were considered to be positive when both sets gave a reliable signal (≤40 CT value) [23].

### Detection and amplification

We used the primers and sequence-specific fluorescence probes were designed tailored to the high conservative region in the COVID-19 genome. The important steps for amplifications were taken place within 50˚C for 15 min, 95˚C for 3 min, followed by 45 cycles of 95˚C for 15 s and 60˚C for 30s. We used RT-PCR instrument, a machine that amplifies and detects RNA. In the case of SARS-CoV-2, the rtqPCR combines the functions of a thermal cycler and a fluorimeter, enabling the process of quantitative PCR. All three cycles were accomplished within an hour and 35 minutes of the reaction being started.

### Hematology tests

Five (5ml) of whole blood samples were collected aseptically from each study subject, fresh (<4 hours from collection) dipotassium EDTA–anticoagulated collected in vacutainer tubes (Becton Dickinson), and run Beckman Coulter DxH 800 Hematology Analyzer (California, USA). With the DxH 800 hematology analyzer, you can utilize accurate data about individual cell size, shape and structure to provide high-quality first-pass results. The machine can analyze and/or test twenty-eight (28) different haematology parameters results.

### Biochemical and enzymatic tests

The collected and coagulated blood from serum separation tubes (SST) were used for enzymatic and biochemical tests for liver and renal function assessment at the chemistry laboratory using Roche Cobas C 501 Chemistry Analyzer at the clinical chemistry laboratory.

This chemistry laboratory was well equipped and integrated with different departments for the diagnostic tests, and it is accredited by the Ethiopian National Accredited Office (ENAO) to perform tests in accordance with the attached scope of accreditation in the field of medical testing in the requirements of International Standard Organization (ISO 15189:2012) since 2019.

### Data quality assurance

All procedures and steps were pre-tested and proper amendments were taken prior to the actual data collection. For laboratory analysis pre-analytical such as identifying of the right patient and laboratory request papers, material preparations etc were done. In the analytical phase, we followed the right laboratory diagnostic and testing procedures and instrument analysis after the samples have been logged into the instrument. We used Westgard rules to assess the validity of the biochemistry values and the rule recommends to tolerating only $1_{2s}$ i.e., one measurement exceeds 2 standard deviations either above or below the mean of the reference range.

In the post-analytical stages, we monitored correctly the result dispatch, and verification of the results of the laboratory work using Standard Operating Procedures (SOPs) of the laboratory.

### Data entry and analysis

Following data cleaning; the sourced data was entered into EPI-Info version 7 using a controlled and programmed data entry format and then exported & analysed using SPSS version 23.0.

A descriptive analysis was done to see the characteristics of the study subjects. Laboratory related continuous variables were categorized using the normal reference range as normal, higher, and lower. We used the chi-square test or Fisher's exact test to assess differences between groups for categorical and dichotomous data. The statistical significance at a 95% CI and p-values ≤ 0.05 was considered statistically significant.

## Ethical issues

Ethical clearance (with ethics approval reference number RM23/3) to carry out this study was obtained from the institutional review board (IRB) from St. Paul's Hospital Millennium Medical College, Addis Ababa, Ethiopia. Written informed consent was taken from each participant. Any information concerning the patients was kept confidential and the specimens collected from the patients were analyzed for the intended purpose only. Positive patient results were communicated between the data collectors and medical doctors who were working at a treatment centre for the better management. Both who have confirmed COVID-19 and suspected patients for infection were managed accordingly by healthcare workers in the isolation and treatment center at St. Paul's Hospital Millennium Medical College, Addis Ababa, Ethiopia.

## Results

### Demographic characteristics

Four hundred thirteen (413) out of 422 calculated total sample patients were participated in the study, yielding a response rate of 97.9%. COVID-19 testing was performed for the 413 patients who had acute respiratory illnesses and were suspected of having COVID-19 disease. More than half of them, n = 244 (59.1%) were males. The median age was 56 years old, with an interquartile range of 25 years old. The majority of patients were married, 277(67.1%), came from urban areas, 346 (83.8%), and had primary and above-primary educational levels 330 (79.9%). Similarly, the majority of patients reported that they were not smoking cigarettes, 397 (96.1%), regularly using alcohol 338 (81.8%), and chewing Khat in 394 (95.4%) of patients. The socio-demographic and related characteristics are displayed in Table 1.

### Clinical characteristics of the participants

Of the 413 patients who presented with acute respiratory illness and went through testing for COVID-19, 250 (60.5%) were confirmed for SARS-CoV-2 infection. The most commonly reported symptoms among suspected patients were shortness of breath, 396 (95.9%) followed by cough, 385 (93.5%) and loss of appetite, 362(87.7%), (Fig 1).

A total of 240 patients (58.1%) were found to have co-morbid chronic health conditions, of whom 138 (57.5%) of them being tested positive for COVID-19 disease. Out of the comorbid chronic health conditions with COVID-19 diseases, hypertension 55 (22.9%) was the most common co-morbid chronic disease, followed by diabetes mellitus, 28 (11.7%), (Fig 2).

### Laboratory results upon presentation

Of the patients tested at the time of presentation, 220 (68.5%) had a higher number of neutrophil counts and 12 (3.7%) had a lower number of neutrophils compared with the normal value. The majority of the patients, 252 (78.3%), had a lower percentage limit of eosinophil count. Differential blood cell count also showed that lymphocyte count was higher in 147 (45.2%) patients. Regarding platelet count, more than one-fourth of patients 97 (29.9%) had higher and 66 (20.4%) had lower platelet numbers at the time of presentation. Despite the fact

**Table 1. Socio-demographic and related characteristics of participants, Addis Ababa, 2022.**

| Characteristics | Frequency | Percent |
|---|---|---|
| Age group | | |
| 18–44 | 118 | 28.6 |
| 45–64 | 168 | 40.7 |
| 65+ | 127 | 30.8 |
| Sex | | |
| Male | 244 | 59.1 |
| Female | 169 | 40.9 |
| Marital status | | |
| Single | 58 | 14.0 |
| Married | 277 | 67.1 |
| Divorce | 16 | 3.9 |
| Widowed | 56 | 13.6 |
| Separated | 6 | 1.5 |
| Residence | | |
| Urban | 346 | 83.8 |
| Rural | 67 | 16.2 |
| Occupation | | |
| Labor worker | 18 | 4.4 |
| Government employee | 79 | 19.1 |
| Unemployed (house wife) | 139 | 33.7 |
| Self-employee | 102 | 24.7 |
| Other | 75 | 18.2 |
| Education | | |
| No formal education | 83 | 20.1 |
| Primary school | 92 | 22.3 |
| Secondary school | 136 | 32.9 |
| Tertiary school | 102 | 24.7 |
| Use of alcohol | | |
| Yes | 75 | 18.2 |
| No | 338 | 81.8 |
| Cigarette Smoking | | |
| Yes | 16 | 3.9 |
| No | 397 | 96.1 |
| Use of Khat | | |
| No | 394 | 95.4 |
| Yes | 19 | 4.6 |
| Hand cleaning | | |
| Yes | 382 | 92.5 |
| No | 31 | 7.5 |
| Method of hand washing | | |
| Soap and Water | 106 | 25.7 |
| Alcohol Based | 8 | 1.9 |
| Antiseptic | 13 | 3.1 |
| Water and antiseptic | 286 | 69.2 |
| Frequency of daily hand washing | | |
| Not at all | 12 | 2.9 |
| 1–3 times | 106 | 25.7 |

(*Continued*)

**Table 1.** (Continued)

| Characteristics | Frequency | Percent |
|---|---|---|
| 4–8 times | 228 | 55.2 |
| 9 or more times | 67 | 16.3 |
| Frequency of rubbing hands using antiseptic | | |
| Not at all | 74 | 17.9 |
| 1–3 times | 78 | 18.9 |
| 4–8 times | 164 | 39.7 |
| 9 or more times | 97 | 23.5 |
| Use of public transport | | |
| No | 90 | 21.8 |
| Yes | 323 | 78.2 |
| Use of Mask | | |
| No | 76 | 18.4 |
| Yes | 337 | 81.6 |
| Aware of 2 meter distance | | |
| No | 43 | 10.4 |
| Yes | 370 | 89.6 |
| Maintain social distance | | |
| Not at all | 90 | 21.8 |
| 25% of the time | 140 | 33.9 |
| 50% of the time | 124 | 30.0 |
| 75% of the time | 59 | 14.3 |

that more than half of the patients 167 (52.4%) had hemoglobin levels in the normal range, a significant number of patients 135 (42.3%) had lower hemoglobin levels. Enzymatic tests like Glutamate Pyruvate Transaminase (GPT) and Aspartate Transaminase (AST) were in normal ranges for the majority of patients during the time of presentation. More than a quarter of patients had a normal range of creatinine level, whereas about one in ten patients had a higher creatinine level. About three in ten patients had higher urea levels in the blood. The main laboratory test results of the patients at the time of presentation, Table 2.

## Comparing socio-demographic and related characteristics

Patients with a positive COVID-19 test result had higher educational status at tertiary school levels than patients with a negative for COVID-19 (71.6% vs 28.4%, p < 0.05) and the only alcohol-based method of hand washing usage was significantly lower among patients with a positive COVID-19 test result (12.5% vs 87.5%, p<0.05), Table 3, than patients with a negative for COVID-19. The prevalence of any comorbidity did not differ significantly between COVID-19 positive and negative patients.

## Comparing clinical profiles of patients with and without COVID-19 disease

Patients with COVID-19 reported a significantly higher proportion of headache (67.3% vs 32.7%, p<0.001); sore throat (72.5% vs 27.5%, p = 0.001); loss of sense of taste (78% vs 22%, p<0.001) and smell (74.4% vs 25.6%, p = 0.002) at a higher rate than COVID-19 negative patients. The presence of fever, cough, and shortness of breath, sneezing, loss of appetite, runny nose, and gastrointestinal symptoms did not differ by COVID-19 status, Table 4.

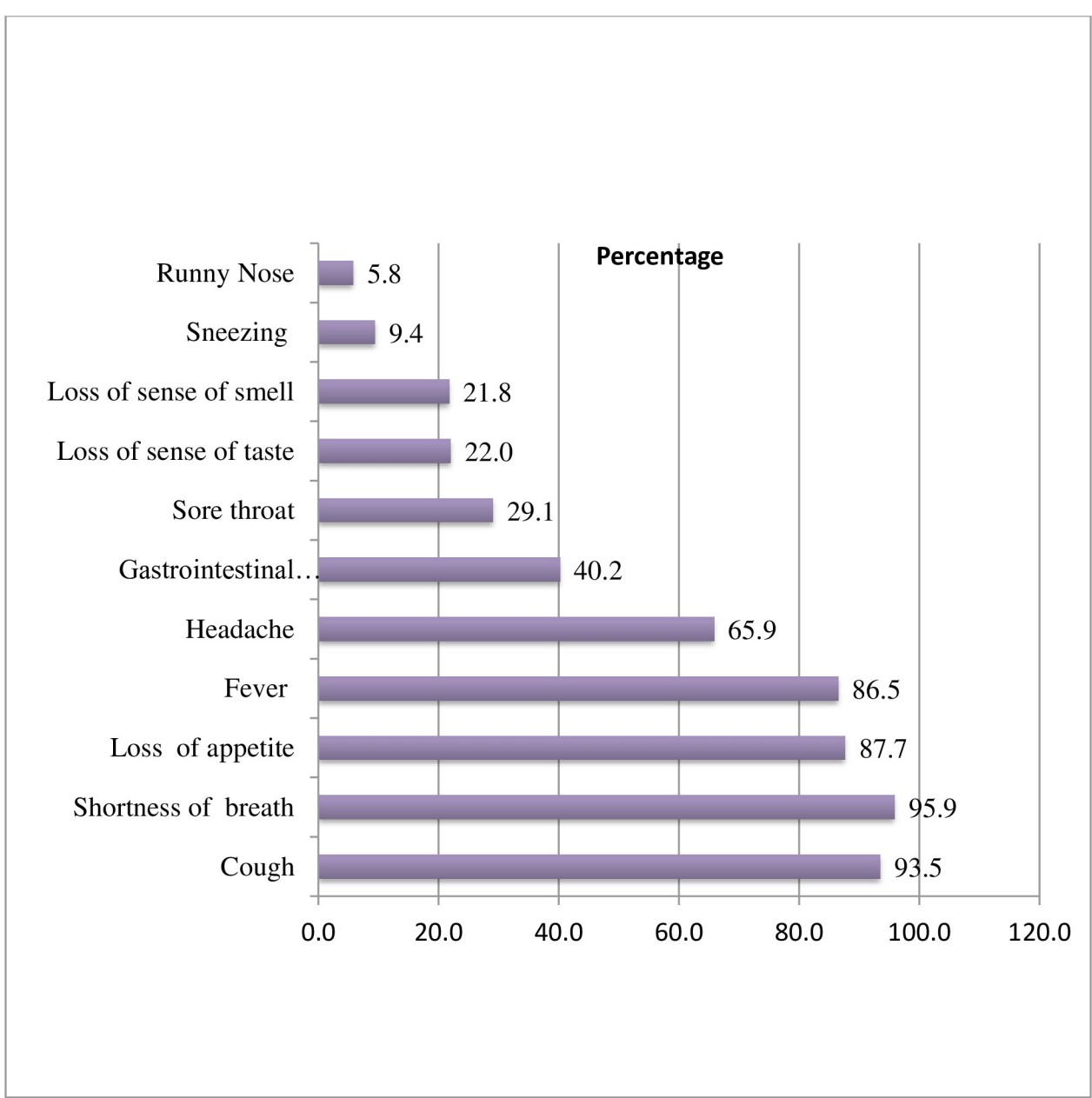

**Fig 1. Commonly reported symptoms of the participants presenting with acute respiratory illness tested for COVID-19 in Addis Ababa, Ethiopia.**

## Laboratory profiles of patients with and without COVID-19 disease

Differential white blood cell count at the time of presentation showed that neutrophil counts with higher than the upper limit. It showed significantly higher in the COVID-19 patients than their counterparts (68.2% vs 31.8%; p<0.05). At the same time, the proportion of COVID-19 patients who had a neutrophil counts lower than the lower limit value, significantly lower (33.3% vs 66.7%; p<0.01). The proportion of COVID-19 patients who had eosinophil counts were lower than the lower limit (69.8% vs 30.2%; p<0.001). It was significantly higher

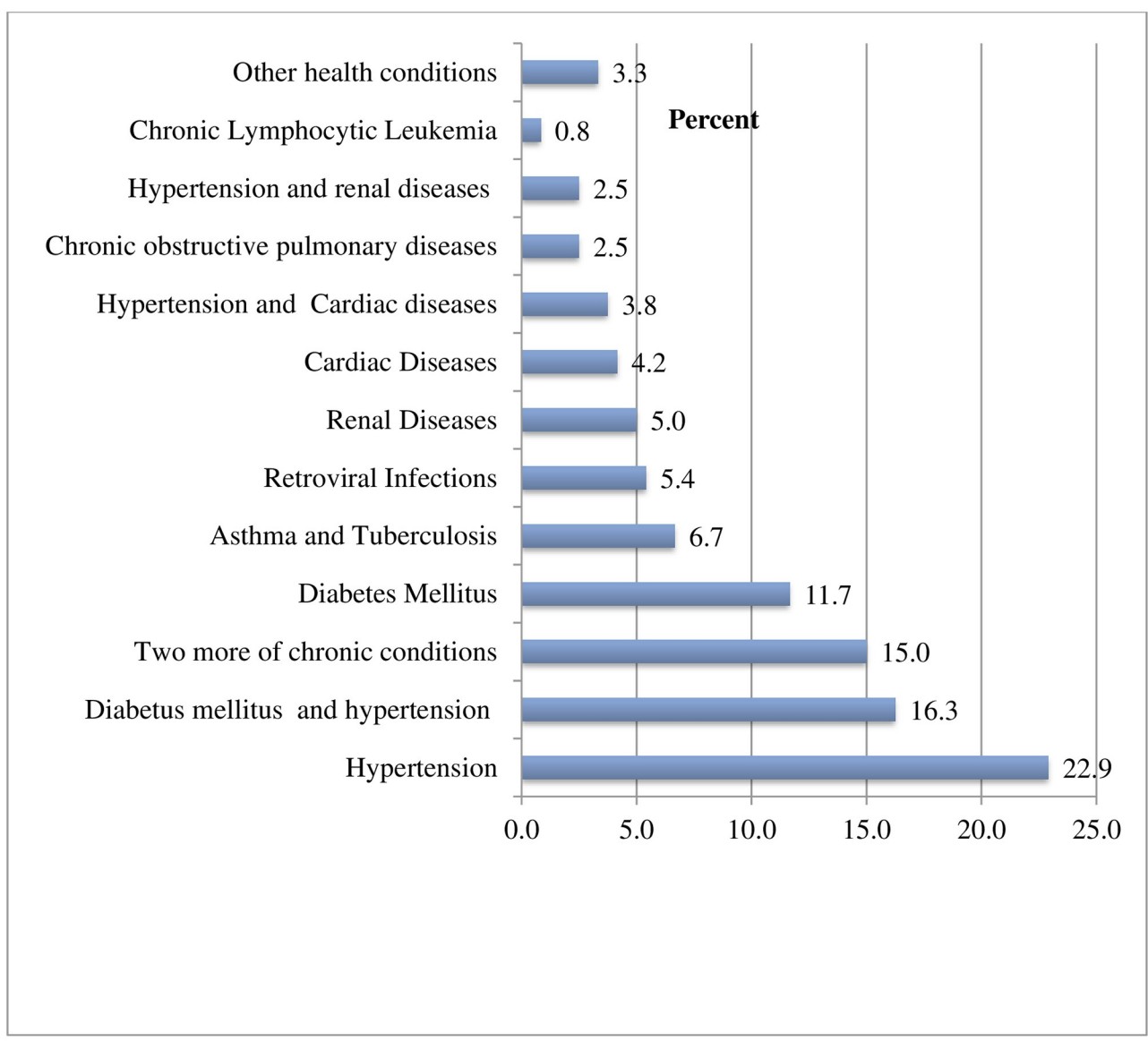

**Fig 2. Underline disease condition status of the participants presenting with acute respiratory illness tested for COVID-19 in Addis Ababa, Ethiopia.**

than patients without COVID-19. Concurrently, COVID-19 patients had eosinophil counts higher than the upper boundary. It was significantly higher than patients without the disease (25.3% vs 72.7%; $p < 0.001$). The percentage of COVID-19 patients with a lower monocyte value was considerably greater than the number of patients without the disease (52.6% vs 47.4%; $p < 0.05$). Patients with COVID-19 have significantly different red blood cell count, hemoglobin, hematocrit level and red cell distribution width (RDW) values as compared to COVID-19 negative patients.

Biochemical profile analysis showed that COVID-19 patients had a significantly higher normal value of creatinine (64.9% vs 35.1%; $p < 0.05$) than the COVID-19 negative patients. And a significantly higher proportion of patients with COVID-19 had a lower value (65.0% vs 35.0%;

**Table 2. Laboratory test results among patients presenting with acute respiratory illness and tested for COVID-19, Addis Ababa, 2022.**

| Hematological tests | Category | Frequency | Percent |
|---|---|---|---|
| White blood cell count (n = 319) | 3.4-10 x10$^9$/L* | 186 | 58.3 |
| Leukopenia | lower than the LL | 14 | 4.4 |
| Leukocytosis | Higher than the UL | 119 | 37.3 |
| Neutrophil count (n = 321) | 4.0-7.5 X 10 $^9$/L* | 89 | 27.7 |
| Neutropenia | Lower than the LL | 12 | 3.7 |
| Neutrophilia | Higher than the UL | 220 | 68.5 |
| Lymphocyte count (n = 325) | 4.0-10 X 10$^9$/L* | 135 | 41.5 |
| Lymphopenia | Lower than the LL | 43 | 13.2 |
| Lymphocytosis | Higher than the UL | 147 | 45.2 |
| Eosinophils (n = 322) | 1-6%* | 59 | 18.3 |
|  | Lower than the LL | 252 | 78.3 |
|  | Higher than the UL | 11 | 3.4 |
| Monocyte (n = 322) | 2-10%* | 249 | 77.3 |
|  | Lower than the LL | 19 | 5.9 |
|  | Higher than the UL | 54 | 16.8 |
| Basophils (n = 318) | 0-1.0%* | 289 | 90.9 |
|  | Higher than the UL | 29 | 9.1 |
| Red blood cell count (n = 306) | 4.3-5.9 x10$^9$/L* | 212 | 69.3 |
|  | Lower than the LL | 83 | 27.1 |
|  | Higher than the UL | 11 | 3.6 |
| Platelet count (n = 324) | 150-450x10$^9$/L* | 161 | 49.7 |
| Thrombocytopenia | Lower than the LL | 66 | 20.4 |
| Thrombocytosis | Higher than the UL | 97 | 29.9 |
| Hemoglobin (n = 319) | 13.6-17.5 g/dL* | 167 | 52.4 |
|  | Lower than the LL | 135 | 42.3 |
|  | Higher than the UL | 17 | 5.3 |
| Hematocrit (n = 319) | 30.7% -47.5%* | 225 | 70.5 |
|  | Lower than the LL | 42 | 13.2 |
|  | Higher than the UL | 52 | 16.3 |
| Mean corpuscular volume (MCV) (n = 306) | 80-94 fl* | 243 | 79.4 |
|  | lower than the LL | 21 | 6.9 |
|  | Higher than the UL | 42 | 13.7 |
| Mean Corpuscular Hemoglobin (MCH) (n = 317) | 27-31 pg per cell* | 196 | 61.8 |
|  | lower than the LL | 28 | 8.8 |
|  | Higher than the UL | 93 | 29.3 |
| Red cell Distribution Width (RDW) (n = 322) | 11.6-14.8%* | 215 | 66.8 |
|  | Higher than the UL | 107 | 33.2 |
| **Biochemical and Enzymatic tests** |  |  |  |
| Creatinine (n = 319) | 0.5-1.2 mg/dL* | 242 | 75.9 |
|  | Lower than the LL | 8 | 2.5 |
|  | Higher than the UL | 69 | 21.6 |
| Urea (n = 339) | 16.6-48.5 mg/dL* | 199 | 58.7 |
|  | Lower than the LL | 40 | 11.8 |
|  | Higher than the UL | 100 | 29.5 |
| Alkaline phosphatase (ALP) (n = 300) | 40-129 IU/L* | 256 | 85.3 |
|  | Lower than the LL | 8 | 2.7 |
|  | Higher than the UL | 36 | 12.0 |

(*Continued*)

**Table 2.** (Continued)

| Hematological tests | Category | Frequency | Percent |
|---|---|---|---|
| Glutamate Pyruvate Transaminase (GPT) (n = 303) | 0-41U/L* | 212 | 70.0 |
| | Higher than the UL | 91 | 30.0 |
| Aspartate transaminase (AST) (n = 303) | 0-44 U/L* | 182 | 60.1 |
| | Higher than the UL | 121 | 39.9 |

* Normal values; LL, lower limit of the normal value; UL, upper limit of the normal value

$p < 0.05$) of blood urea. The percentage of patients with COVID-19 with a normal value of alkaline phosphatase (ALP) was higher than patients without the disease (66.8% vs 33.2%; $p < 0.05$).

However, there was no significant difference observed in haematological tests such as white blood cell counts, platelet counts, lymphocytes and basophils and enzymatic test values like glutamate pyruvate transaminase (GPT) and aspartate transaminase (AST), mean corpuscular volume (MCV) and mean corpuscular hemoglobin (MCH) by COVID-19 status as observed in Table 5.

## Discussion

Despite many studies describing clinical features of patients with COVID-19 [1–4, 16, 18], few have directly compared the clinical presentation, laboratory diagnosis, and other important characteristics of patients [10–13, 19, 20]. The majority of studies comparing the clinical features of COVID-19 patients with acute respiratory illnesses (ARI) were from developed countries which may not reflect the context of developing countries like Ethiopia. Hence, the current study compares epidemiological, behavioral and lifestyle, clinical, laboratory and other variables among patients with confirmed severe acute respiratory syndrome (SARS-CoV-2) infection and suspected for COVID-19 without disease.

This study found a higher proportion of COVID-19 patients among patients with acute respiratory illnesses at the time of the study using RT- PCR testing. One explanation for the high prevalence of the disease among participants could be due to more than half of the participants n = 138(57.7%) were having comorbid disease. This justification was supported by other studies, comorbidity may increase the chance of contracting ARIs or COVID-19 disease [24, 25]. The other reason for this high proportion of disease also may be due to participants were recruited from the main referral institution. However, the testing of suspected patients in this study could contribute largely to mitigating the spread of infection within a hospital and the community at large as described by previous studies [26, 27], and especially if rapid testing is implemented, significantly minimize the impact in low-income nations [28].

This study clinically revealed that COVID-19 positive patients had a significantly greater proportion of headache, sore throat, and loss of smell and taste than COVID-19 negative patients. The current findings on sore throat symptoms are congruent with another study, that compared COVID-19 patients' clinical symptoms to other acute respiratory symptoms [19, 29–31]. Our findings, however, contradict those of a large retrospective cohort study comparing clinical aspects of COVID-19 patients with influenza [32] and other respiratory diseases [33], which found that sore throat was not substantially more common in COVID-19 patients. However, the former study [32] varies from the current study that clinical symptoms of COVID-19 were compared to clinical symptoms of influenza using a propensity matching technique at different times, which could introduce context-dependent biases because low-

**Table 3. COVID-19 test results for the participants presenting with acute respiratory illness in Addis Ababa, 2022.**

| Variables | COVID-19 Positive (n = 250) | COVID-19 Negative (n = 163) | P-Values |
|---|---|---|---|
| Age Category | | | |
| 18–44 | 62(52.5%) | 56(47.5%) | 0.099 |
| 45–64 | 109(64.9%) | 59(35.1%) | |
| 65+ | 79(62.2%) | 48(37.8%) | |
| Marital status | | | |
| Single | 29 (50%) | 29 (50%) | 0.357 |
| Married | 173(62.5%) | 104(37.5) | |
| Separated/Divorced | 13 (59.1%) | 9(40.9%) | |
| Widowed | 35(62.5) | 21(37.5) | |
| Sex | | | |
| Male | 154(63.1%) | 90(36.9%) | 0.197 |
| Female | 96(56.8%) | 73(43.2%) | |
| Resident | | | |
| Urban | 208(60.1%) | 138(39.9%) | 0.694 |
| Rural | 42(62.7%) | 25(37.3%) | |
| Occupational status** | | | |
| Labor | 9(50%) | 9(50%) | 0.614 |
| Government | 53(67.0%) | 26(33.0%) | |
| House wife | 77(55.4%) | 62(44.6%) | |
| Self-employee | 65(63.7) | 37(36.3%) | |
| Others | 46(61.3%) | 29(38.7%) | |
| Educational Status | | | |
| No formal education | 49(59.0%) | 34(41.0%) | 0.033 |
| Primary school | 47(51.1%) | 45(48.3%) | |
| Secondary school | 81(59.6%) | 55(40.4%) | |
| Tertiary school | 73(71.6%) | 29(28.4%)* | |
| Alcohol consumption | | | |
| Yes | 39(52.0%) | 36(48.0%) | 0.095 |
| No | 211(62.4%) | 127(37.6%) | |
| Smoking | | | |
| Yes | 241(60.7%) | 156(39.3%) | 0.721 |
| No | 9(56.3) | 7(43.8%) | |
| Khat chewing | | | |
| Yes | 14(73.7%) | 5(26.3%) | 0.230 |
| No | 236 (59.9%) | 158(40.1%) | |
| Clean hand | | | |
| Yes | 233(61.0%) | 149(39.0%) | 0.500 |
| No | 17(54.8%) | 14(45.2%) | |
| Method of hand washing** | | | |
| Soap and Water | 67(63.2%) | 39(36.8%) | 0.048 |
| Alcohol rubbing | 1(12.5%) | 7(87.5%)* | |
| Antiseptic | 8(61.5%) | 5(38.5%) | |
| Water and antiseptic | 174(60.8%) | 112(39.2%) | |
| Daily hand washing** | | | |
| Not at all | 8(66.7%) | 4(33.3%) | 0.065 |
| 1–3 times | 59(55.7%) | 47(44.3%) | |
| 4–8 times | 140(61.4%) | 88(38.6%) | |

*(Continued)*

**Table 3.** (Continued)

| Variables | COVID-19 Positive (n = 250) | COVID-19 Negative (n = 163) | P-Values |
|---|---|---|---|
| 9 or more times | 43(64.2%) | 24(35.8%) | |
| Daily rubbing hands | | | |
| Not at all | 43(58.1%) | 31(41.9%) | 0.453 |
| 1–3 times | 48(61.5%) | 30(38.5%) | |
| 4–8 times | 94(57.3%) | 70(42.7%) | |
| 9 or more times | 65(67.0%) | 32(33%) | |
| Use of public transport | | | |
| Yes | 199(61.6%) | 124(38.4%) | 0.396 |
| No | 51(56.7%) | 39(43.3%) | |
| Use of mask | | | |
| No | 43(56.6%) | 33(43.4%) | 0.435 |
| Yes | 207(61.4%) | 130(38.6%) | |
| Awareness of 2m distance | | | |
| No | 22(51.2%) | 21(48.8%) | 0.184 |
| Yes | 228(61.6%) | 142(38.4%) | |
| Social distance | | | |
| Not at all | 59(65.6%) | 31(34.4%) | 0.385 |
| 25% time | 82(58.6%) | 58(41.4%) | |
| 50% time | 78(62.9%) | 46(37.1%) | |
| 75% time | 31(52.5%) | 28(47.5%) | |
| Comorbidity | | | |
| No | 112(64.7%) | 61(35.3%) | 0.805 |
| Yes | 138(57.7%) | 102(42.5%) | |

* denotes a subset of COVID-19 categories whose proportions do differ significantly from each other at the 0.05 level

**Fisher's exact test of significance

resource nations have diverse contexts. The current study agrees with other studies which showed headache [33] and loss of smell [31, 33–35] were significantly higher in COVID-19 patients when compared to other respiratory symptoms.

The clinical symptoms of fever, cough, shortness of breath, loss of appetite, runny nose and gastrointestinal symptoms in the current study did not differ by COVID-19 status. However, comparative studies showed that fever [20, 29], cough [35–37], dyspnea [20, 30] loss of appetite and loss of taste [31] were significantly in higher proportion among COVID-19 patients than without COVID-19. The reason for the variation of symptom profiles between the current study and other studies can be attributed to the difference in the geographic location and the prevalence of underlying chronic disease diseases. Such variation may be accounted for the difference in reporting symptoms as mentioned by one global study [38]. The symptoms at presentation cannot be fully explained because of differences in chronic disease profiles across countries. Taking such symptom variations into consideration, it would give important insights for clinical diagnosis and treatments as well as public health messages that may be individualized based on the countries or geographical locations. This may suggest the need for further contextual and large-scale studies to ensure the predictive clinical symptoms for specific populations.

Of the socio-demographic and epidemiological factors, only educational status and using the alcohol-based method of hand washing (hand rubbing with alcohol) were statistically

**Table 4. Symptoms of the participants among COVID-19 positive and negative individuals presenting with acute respiratory illness and tested for COVID-19, Addis Ababa, 2022.**

| Symptoms | COVID-19 Positive (n = 250) | COVID-19 Negative (n = 163) | P-Values |
|---|---|---|---|
| Cough | | | |
| No | 12(44.4%) | 15(55.6%) | 0.077 |
| Yes | 238(61.7%) | 148(38.3%) | |
| Sneezing | | | |
| No | 226(60.4%) | 148(39.6%) | 0.893 |
| Yes | 24(61.5%) | 15(38.5%) | |
| Shortness of breath | | | |
| No | 99(92.5%) | 8(7.5%) | 0.001 |
| Yes | 201(56.5%) | 155(43.5%) | |
| Headache | | | |
| No | 67(47.5%) | 74(52.5%) | 0.001 |
| Yes | 183(67.3%) | 89(32.7%) | |
| Sore throat | | | |
| No | 163(55.6%) | 130(44.4%) | 0.001 |
| Yes | 87(72.5%) | 33(27.5%) | |
| Runny nose | | | |
| No | 233(59.9%) | 156(40.1%) | 0.287 |
| Yes | 17(70.8%) | 7(29.2%) | |
| Gastrointestinal symptom | | | |
| No | 143(57.9%) | 104(42.1%) | 0.181 |
| Yes | 107(64.5%) | 59(35.5%) | |
| Loss of sense of smell | | | |
| No | 183(56.7%) | 140(43.3%) | 0.002 |
| Yes | 67(74.4%) | 23(25.6%) | |
| Loss of sense of taste | | | |
| No | 179(55.6%) | 143(44.4%) | 0.001 |
| Yes | 71(78.0%) | 20(22.0%) | |
| Loss of appetite | | | |
| No | 25(49.0%) | 26(51.0%) | 0.072 |
| Yes | 225(62.2%) | 137(37.8%) | |
| Fever | | | |
| No | 57(63.3%) | 33 (36.7) | 0.539 |
| Yes | 193(59.8%) | 130 (40.2%) | |

significant factors of COVID-19 positivity. Based on the finding, those who have an educational level of tertiary level had higher risk of a positive COVID-19 test result. This could be due to the fact that these groups of people are more likely to congregate in places like the workplace, which increases the risk of transmission. Using the alcohol-based method of hand washing reduced the chance of getting tested positive in the study. This could be owing to the fact that washing hands with disinfectants inhibit the disease transmission.

However, we found no significant differences between patients with and without COVID-19 with regard to epidemiological factors such as; the use of masks, use of public transport, and risk factors; smoking, comorbidity, alcohol consumption and demographic factors like age. Similarly, most of hematological and enzymatic testing such as GPT and AST revealed no significant differences in the COVID-19 status. However that, some of laboratory findings

**Table 5. Laboratory test results among COVID-19 positive and COVID-19 negative participants presenting with acute respiratory illness, Addis Ababa, 2022.**

| Hematological tests | Values | COVID-19 Positive | COVID-19 Negative | P-Values |
|---|---|---|---|---|
| White blood cell count (n = 319)* | 3.4-10 x10$^9$/L | 116(62.4%) | 70(37.6%) | 0.249 |
| Leukopenia | lower than the LL | 6(42.9%) | 8(57.1%) | |
| Leukocytosis | Higher than the UL | 78(65.5%) | 41(34.5%) | |
| Neutrophil count (n = 321) ** | 4.0-7.5 x 10$^9$/L | 46(51.7%) | 43(48.3%) | 0.001 |
| Neutropenia | lower than the LL | 4(33.3%) | 8(66.7%) | |
| Neutrophilia | Higher than the UL | 150(68.2%) | 70(31.8%) | |
| Lymphocyte count (n = 325) * | 4.0-10 x10$^9$/L | 85(63.0%) | 50(37.0%) | 0.645 |
| Lymphopenia | lower than the LL | 29(67.4%) | 14(32.6%) | |
| Lymphocytosis | Higher than the UL | 88(59.9%) | 59(40.1%) | |
| Eosinophils (n = 322) * | 1-6% | 21(35.6%) | 38(64.4%) | 0.001 |
| | lower than the LL | 176(69.8%) | 76(30.2%) | |
| | Higher than the UL | 3(25.3%) | 8(72.7%) | |
| Monocyte (n = 322) * | 2-10% | 165(66.3%) | 84(33.7%) | 0.016 |
| | lower than the LL | 10(52.6%) | 9(47.4% | |
| | Higher than the UL | 25(46.3%) | 29(53.7%) | |
| Basophils (n = 318) * | 0-1.0% | 182(63.0%) | 107(37.0%) | 0.121 |
| | Higher than the UL | 14(48.3%) | 15(51.7%) | |
| | Higher than the UL | 25(46.3%) | 29(53.7%) | |
| Red blood cell count (n = 306) ** | 4.3-5.9 x 10$^9$/L | 150(70.8%) | 62(29.2%) | 0.001 |
| | lower than the LL | 39(47.0%) | 44(53.0%) | |
| | Higher than the UL | 6(54.5%) | 5(45.5%) | |
| Platelet count (n = 324) * | 150-450 X 10$^9$/L | 104(64.6%) | 57(35.4%) | 0.546 |
| Thrombocytopenia | lower than the LL | 41(62.1%) | 25(37.9%) | |
| Thrombocytosis | Higher than the UL | 56(57.7%) | 41(42.3%) | |
| Hemoglobin (n = 319) * | 13.6-17.5 g/dL | 121(72.5%) | 46(27.5%) | 0.001 |
| | lower than the LL | 67(49.6%) | 68(50.4%) | |
| | Higher than the UL | 12(70.6) | 5(29.4%) | |
| Hematocrit (n = 319) * | 30.7% -47.5% | 150(66.7%) | 75(33.3%) | 0.001 |
| | lower than the LL | 14(33.3%) | 28(66.7% | |
| | Higher than the UL | 36(69.2%) | 16(30.8%) | |
| Mean corpuscular volume (MCV) (n = 306) * | 80-94 fl | 156(64.2%) | 87(35.8%) | 0.713 |
| | Lower than the LL | 12(57.1%) | 9(42.9%) | |
| | Higher than the UL | 25(59.5%) | 17(40.5%) | |
| Mean corpuscular hemoglobin (MCH) (n = 317) * | 27-31 pg | 120(61.2%) | 76(38.8%) | 0.363 |
| | Lower than the lower limit | 16(57.1%) | 12(42.9% | |
| | Higher than the UL | 64(68.8%) | 29(31.2% | |
| Red cell distribution width (RDW) (n = 322) * | 11.6-14.8% | 159(74.0%) | 56(26.0% | 0.001 |
| | Higher than the UL | 41(38.3%) | 66(61.7) | |
| **Clinical chemistry tests** | | | | |
| Creatinine (n = 319) ** | 0.5-1.2 mg/dL | 157(64.9%) | 85(35.1%) | 0.047 |
| | lower than the LL | 4(50.0%) | 4(50.5%) | |
| | Higher than the UL | 34(49.3%) | 35(50.7%) | |
| Urea (n = 339) * | 16.6-48.5 mg/dL | 131(65.8%) | 68(34.2) | 0.026 |
| | lower than the LL | 26(65.0%) | 14(35.0%) | |
| | Higher than the UL | 50(50.0%) | 50(50.0%) | |

(*Continued*)

**Table 5.** (Continued)

| Hematological tests | Values | COVID-19 Positive | COVID-19 Negative | P-Values |
|---|---|---|---|---|
| ALP (n = 300)** | 40-129 IU/L | 171 (66.8%) | 85(33.2%) | 0.046 |
| | lower than the LL | 4(50.0%) | 4(50.0%) | |
| | Higher than the UL | 17(47.2%) | 19(52.8%) | |
| G PT (n = 303) * | 0-41U/L | 133(62.7%) | 79(37.3%) | 0.475 |
| | Higher than the UL | 61(67.0%) | 30(33.0%) | |
| AST (n = 303)* | 0-44 U/L | 112 (61.5%) | 70(38.5%) | 0.268 |
| | Higher than the UL | 82(67.8%) | 39(32.2%) | |

* Normal values; LL, the Lower limit of the normal value; UL, the upper limit of the normal value;

** Fisher exact test

COVID-19 patients like; neutrophil, eosinophil, and monocyte from white blood cells count (WBC); red blood cells (RBCs), hemoglobin (Hgb) and hematocrite (Hct) level and red cell distribution width (RDW) from red blood cells count; creatinine (Cr) and blood urea nitrogen (BUN) from renal function test; only alkaline phosphatase from liver function test were significantly different compared to COVID-19 negative patients.

The current study inconsistent with the findings of other comparative studies [13, 19, 20] in that lymphopenia was more common and cardinal laboratory finding in COVID-19 patients than in patients without the disease [37, 39]. Although many studies have shown that a low lymphocyte count may predict the severity of the COVID-19 [40, 41], in the current study low lymphocyte counts did not predict the disease status, and this may be because of co-morbidities that could be associated with an increased risk of lymphopenia, such as hypertension and diabetes [42] which are really common co-morbidities in this study. As a result, significant difference may not be observed among patients with and without COVID-19 as the comparability of marked systemic increase of inflammatory mediators and cytokines which contribute to comorbid and COVID-19-associated lymphopenia.

## Limitation of the study

As a limitation, the study was conducted at the referral and specialized hospital in Addis Ababa, Ethiopia serving for patients various corners of the country. Additionally, participants were presenting any acute respiratory illness of suspected for COVID-19 disease. Hence, the proportion of getting such kind of diseased patients may result exaggerated proportion. However, as strength, it has a comprehensive content deal with epidemiology of the disease, clinical and different of laboratory parameters to indicate between COVID-19 confirmed and diseased free patients, which will be informative for other resource limited country.

## Conclusion

This study identified that COVID-19 patients had a significantly higher proportion of headache, sore throat, loss of smell and taste when compared to COVID-19 negative patients. The study also found significant differences in differential blood cell counts of neutrophil, eosinophil, and monocyte, RBCs, Hgb, Hct, RDW; biochemical renal function tests like creatinine and BUN; and enzymatic tests of ALP at the time of presentation between COVID-19 confirmed patients and without the disease.

These findings also suggest the need to substantially consider for educated people, and using of alcohol rubbing during hand washing; the clinical symptoms of headache, sore throat, loss of sense of smell and taste may also be useful as a potential clinical diagnostic symptoms for early screening and testing for the country's context. Giving an attention for those clinical symptoms and laboratory profiles, it may help for early identification and diagnosis of COVID-19 among suspected patients in the limited resources, like on the condition of scarcity of PCR machine, particularly in our setting.

## Supporting information

**S1 Checklist. STROBE statement—Checklist of items that should be included in reports of observational studies.**
(DOCX)

## Acknowledgments

We thank the SPHMMC COVID-19 isolation, testing and treatment centres staffs for supporting, testing and clinical investigations. We also thank patients who participated in the study.

## Author Contributions

**Conceptualization:** Addisu Gize, Melkayehu Kassa, Solomon Ali, Yosef Tadesse, Bereket Fantahun, Aman Yesuf.

**Data curation:** Addisu Gize, Melkayehu Kassa, Solomon Ali, Aman Yesuf.

**Formal analysis:** Addisu Gize, Yosef Tadesse, Aman Yesuf.

**Funding acquisition:** Addisu Gize.

**Investigation:** Addisu Gize.

**Methodology:** Aman Yesuf.

**Project administration:** Solomon Ali, Bereket Fantahun.

**Supervision:** Addisu Gize, Melkayehu Kassa, Yosef Tadesse, Bereket Fantahun.

**Writing – original draft:** Yitagesu Habtu.

**Writing – review & editing:** Addisu Gize, Solomon Ali, Yosef Tadesse, Bereket Fantahun, Yitagesu Habtu.

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
