## [Decision Letter · Decision Letter 0]

2 Jun 2023

PONE-D-23-08723Epidemiological, clinical and laboratory profile of patients presenting with severe acute respiratory syndrome (SARS-CoV-2) in Ethiopia: Comparative cross-sectional studyPLOS ONE

Dear Dr. Gize,

Thank you for submitting your manuscript to PLOS ONE. After careful consideration, we feel that it has merit but does not fully meet PLOS ONE’s publication criteria as it currently stands. Therefore, we invite you to submit a revised version of the manuscript that addresses the points raised during the review process.

We look forward to receiving your revised manuscript.

Kind regards,

Enoch Aninagyei, PhD

Academic Editor

PLOS ONE

Journal Requirements:

   "This research was funded by the St. Paul’s Hospital Millennium Medical College. The funder had no role of the in designing of the study, collection, analysis, and interpretation of data."

Reviewers' comments:

Reviewer's Responses to Questions

**Comments to the Author**

1. Is the manuscript technically sound, and do the data support the conclusions?

Reviewer #1: Partly

Reviewer #2: Yes

2. Has the statistical analysis been performed appropriately and rigorously? 

Reviewer #1: Yes

Reviewer #2: No

3. Have the authors made all data underlying the findings in their manuscript fully available?

Reviewer #1: Yes

Reviewer #2: Yes

4. Is the manuscript presented in an intelligible fashion and written in standard English?

Reviewer #1: No

Reviewer #2: Yes

5. Review Comments to the Author

Reviewer #1: Good attempt to review and report on COVID-19 cases that were seen in your facility. However, take note of the following observations:

1. The entire manuscript needs grammatical correction. I recommend they consult an editor to review the manuscript and correct the writings before submission.

Results

1. The authors indicated a sample of 413 respondents, however, on Table 2, all variables reported had different sample size or denominator. How is this possible for haematological parameters run at the same time?

2. The authors mentioned co-morbidities that is hypertension and diabetes. These two diseases may not cause the elevation of the haematological parameters like the Neutrophils and enzymematic markers such as ALP, however, other disease conditions like liver disease and other chronic inflammatory conditions may present with similar elevations. Therefore, the authors must indicate which specific chronic conditions or metabolic diseases that were ruled out in the study apart the hypertension and diabetes.

Discussion

1. The discussion is not cogent, there are contradictory statements and unfounded assumptions.

2. On paragraph 2, the assumption made on the presence of comorbidity and ARI increases the positivity of COVID-19 should be further explain since the assumption does not clearly indicate specifically true positivity reaction or false positivity reaction.

Conclusion

The conclusion is not precise and concise, rather a repetition of the results and aspects of the discussion. Consider rewording the paragraph 2 as the conclusion of the study. However, the clinical diagnostics symptoms reported is a general knowledge that has been reported by several studies.

Given your findings, are you recommending that will diagnosis and treatment of COVID-19 patients could be done based on symptoms rather than PCR test?

Reviewer #2: This article aimed to compare epidemiological, clinical, and laboratory profiles of patients presenting with acute respiratory syndrome illness in Addis Ababa Ethiopia. The specific research question is whether the epidemiological, clinical, and laboratory profile data of COVID-19 patients managed in low-resource settings might be quite different from findings from developed countries.

The study reports the results of N=413 patients who had acute respiratory illnesses and were suspected of having COVID-19 disease. A total of 250 (60.5%) patients tested positive for COVID-19 disease. COVID-19 patients were less likely to use an alcohol-based method of hand washing, and a higher proportion of headaches, sore throat, and loss of sense of taste.

The study further found that Neutrophil, creatinine, and alkaline phosphatase were significantly higher in COVID-19 patients.

The strength of the study was that a lot of variables were collected, and a well-written manuscript. In addition, the study aims at comparing epidemiological, clinical, and laboratory profiles of patients presenting with acute respiratory syndrome illnesses. This research team is in a unique position to address this question. The results are thus of potentially great interest, but there are some issues with the manuscript that need to be addressed.

Major Comments

Line 282-283: Authors should run a logistic regression to test for association.

Minor Comments

Abstract

Line 33: delete ‘was’ (the study aimed to compare).

Line 34: Add ‘of’ (clinical and laboratory profiles of patients).

Line 38: Add ‘a’ (Using a structured questionnaire)

Line 41: Replace (to see their laboratory profiles) with ‘for laboratory profile analysis’.

Line 45: delete ‘were’ (participants tested positive)

Methods

Line 111: what about those who are 18 years and below?

Line 113: A suspected case is based on meeting one or more of either clinical or epidemiological link criteria. It is not clear why, according to the authors, patients were tested irrespective of whether they met the criteria or not.

Line 117: The sentence should be paraphrased.

Line 134: Per the calculation, the minimum sample size must be 423.

Line 142: delete ‘an.’

Line 149: This statement contradicts that of line 113. The authors must clarify this.

Line 156: Provide a reference for the protocol or the standard operating procedure.

Line 182: With the biochemical and enzymatic tests, the blood was collected into which tube since an inappropriate tube can lead to wrong results. E.g., sodium fluoride (NaF) tubes and serum separation tubes (SST)

Line 192-193: Authors should state exactly what was done to prevent or minimize pre-analytical, analytical, and post-analytical errors. Specimen collection, handling, storage including temp, Assays used, quality control testing, and reliability of assays.

Results

Line 211: Indicate the figure.

Line 216: Authors should re-categorize the groups i.e., Occupation. I think some health professionals, as well as drivers, are Government employees.

-Education: The word illiterate is too offensive. Replace with no formal education.

College and above: replace with tertiary.

Line 231: Authors must be consistent in reporting. Just use the percentages

Line 260: Use a ‘differential blood count.’

Discussion

Line 294-295: It is not clear which respiratory tract infections apart from Covid-19, that, the authors compare the clinical, laboratory, and other variables with.

Line 296-297: the study site is a referral and treatment center. Most patients are likely to test positive for Covid-19. Moreover, some might be severely ill and have received some form of treatment before being referred to the facility. This can lead to bias in the study.

6. PLOS authors have the option to publish the peer review history of their article (what does this mean?). If published, this will include your full peer review and any attached files.

Reviewer #1: **Yes: **Comfort Dede Tetteh

Reviewer #2: No

---

## [Author Response · Author response to Decision Letter 0]

18 Jun 2023

PONE-D-23-08723

Epidemiological, clinical and laboratory profile of patients presenting with severe acute respiratory syndrome (SARS-CoV-2) in Ethiopia: Comparative cross-sectional study

Dear Dr. Gize,

Thank you for submitting your manuscript to PLOS ONE. After careful consideration, we feel that it has merit but does not fully meet PLOS ONE’s publication criteria as it currently stands. Therefore, we invite you to submit a revised version of the manuscript that addresses the points raised during the review process.

We look forward to receiving your revised manuscript.

Kind regards,

Enoch Aninagyei, PhD

Academic Editor

PLOS ONE

Response: Dear Enoch Aninagyei, PhD, Academic Editor, PLOS ONE

We authors would like to thank for your invitation to submit our revised version of the manuscript, PONE-D-23-08723, 

Epidemiological, clinical and laboratory profile of patients presenting with severe acute respiratory syndrome (SARS-CoV-2) in Ethiopia: Comparative cross-sectional study, that addresses raised points during the review process.

 We included a point-by-point response within the 'Response to Reviewers' box in the submission system and modifications are highlighted in the track changes of the original manuscript, and we uploaded also the clean or unmarked version of our revised paper without tracked changes.

Thank you

S. No Points raised by editors and /or reviewers’ Responses to points

1 Please ensure that your manuscript meets PLOS ONE's style requirements, including those for file naming. In accordance with the editor’s concern, we have now

Confirmed that our revised manuscript meets PLOS ONE’s style requirements.

2 Please note that funding information should not appear in any section or other areas of your manuscript We confirmed that funding information is not appeared in any section of the revised version.

3 We note that the grant information you provided in the ‘Funding Information’ and ‘Financial Disclosure’ sections do not match. 

When you resubmit, please ensure that you provide the correct grant numbers for the awards you received for your study in the ‘Funding Information’ section. We have now ensured you that the correct grant number which we received in the funding information as “This research was funded by the St. Paul’s Hospital Millennium Medical College. The funder had no role of the in designing of the study, collection, analysis, and interpretation of data. 

4 Your ethics statement should only appear in the Methods section of your manuscript. If your ethics statement is written in any section besides the Methods, please move it to the Methods section and delete it from any other section. Please ensure that your ethics statement is included in your manuscript, as the ethics statement entered into the online submission form will not be published alongside your manuscript. We have corrected that, our ethics statement of the revised version of the manuscript is written only in the methods section. 

5 Please include captions for your Supporting Information files at the end of your manuscript, and update any in-text citations to match accordingly. Please see our Supporting Information guidelines for more information: http://journals.plos.org/plosone/s/supporting-information. Thank you for your concern. We have now included the three captions of our supporting Information file at the end of our revised manuscript.

 Reviewer's Responses to Questions

Comments to the Author

1. Is the manuscript technically sound, and do the data support the conclusions?

Reviewer #1: Partly

Reviewer #2: Yes 

We would like to thank all reviewers for their critical evaluation and giving such kind of evaluation from our manuscript.

 2. Has the statistical analysis been performed appropriately and rigorously? 

Reviewer #1: Yes

Reviewer #2: No With all due respect to the reviewer #2, we believe that this point is not correct. The statistical analysis has been performed appropriately and rigorously by our public health professionals (Epidemiologist). 

 3. Have the authors made all data underlying the findings in their manuscript fully available?

Reviewer #1: Yes

Reviewer #2: Yes Again, authors would like to thank both reviewers for their critical judgment on our manuscript data. 

 4. Is the manuscript presented in an intelligible fashion and written in standard English?

Reviewer #1: No

Reviewer #2: Yes Reviewer #1 concern is well taken, and any typographical or grammatical errors on our manuscript has corrected in the revised version of manuscript by experts.

 5. Review Comments to the Author

 Possible responses for reviewers comment and concerns 

 Reviewer #1: Good attempt to review and report on COVID-19 cases that were seen in your facility. However, take note of the following observations:

1. The entire manuscript needs grammatical correction. I recommend they consult an editor to review the manuscript and correct the writings before submission. Thank you very much for your constructive comment. The comment is accepted. As far as possible the revised version of our manuscript grammar is edited by the best English language speaker before submission. 

 Results

1. The authors indicated a sample of 413 respondents, however, on Table 2, all variables reported had different sample size or denominator. How is this possible for haematological parameters run at the same time? It is true that the total sample size was 413 as indicated in the socio-demographic table. However, with different reason, all participants might not give their clinical sample to the stated objective of study. Written informed consent was taken, and patients were participated voluntarily. So, we did not urge participants who refuse to give all required clinical samples. 

 2. The authors mentioned co-morbidities that is hypertension and diabetes. These two diseases may not cause the elevation of the haematological parameters like the Neutrophils and enzymematic markers such as ALP, however, other disease conditions like liver disease and other chronic inflammatory conditions may present with similar elevations. Therefore, the authors must indicate which specific chronic conditions or metabolic diseases that were ruled out in the study apart the hypertension and diabetes.

 We agree that this is an important area that requires further research, and may not our objective. However, in our study, we have only described participants’ chronic or underlined disease condition, in figure 2.

 Discussion

1. The discussion is not cogent; there are contradictory statements and unfounded assumptions.

2. On paragraph 2, the assumption made on the presence of comorbidity and ARI increases the positivity of COVID-19 should be further explain since the assumption does not clearly indicate specifically true positivity reaction or false positivity reaction. We acknowledge that our manuscript might not have been indicated specifically true positivity reaction or false positivity reaction. But we have described about our participants disease condition saying that “total of 240 patients (58.1%) were found to have co-morbid chronic health conditions, of whom 138 (57.5%) of them being tested positive for COVID-19 disease”, stated in COVID-19 status: Signs and symptoms section of our manuscript.

 Conclusion

The conclusion is not precise and concise, rather a repetition of the results and aspects of the discussion. Consider rewording the paragraph 2 as the conclusion of the study. However, the clinical diagnostics symptoms reported is a general knowledge that has been reported by several studies.

Given your findings, are you recommending that will diagnosis and treatment of COVID-19 patients could be done based on symptoms rather than PCR test? 

The comment is accepted. We have modified the conclusion section on paragraph 2 and the entire section of conclusion based on your constructive comment. 

 Reviewer #2: This article aimed to compare epidemiological, clinical, and laboratory profiles of patients presenting with acute respiratory syndrome illness in Addis Ababa Ethiopia. The specific research question is whether the epidemiological, clinical, and laboratory profile data of COVID-19 patients managed in low-resource settings might be quite different from findings from developed countries. The comment/concern is accepted. There is the scarcity of literature in resource limited country, and as you highlighted the specific research question is whether the epidemiological, clinical, and laboratory profile data of COVID-19 patients managed in low-resource settings might be quite different from findings from developed countries.

 The study reports the results of N=413 patients who had acute respiratory illnesses and were suspected of having COVID-19 disease. A total of 250 (60.5%) patients tested positive for COVID-19 disease. COVID-19 patients were less likely to use an alcohol-based method of hand washing, and a higher proportion of headaches, sore throat, and loss of sense of taste.

The study further found that Neutrophil, creatinine, and alkaline phosphatase were significantly higher in COVID-19 patients.

The strength of the study was that a lot of variables were collected, and a well-written manuscript. In addition, the study aims at comparing epidemiological, clinical, and laboratory profiles of patients presenting with acute respiratory syndrome illnesses. This research team is in a unique position to address this question. The results are thus of potentially great interest, but there are some issues with the manuscript that need to be addressed. It is true that our finding stated that COVID-19 patients were less likely to use an alcohol-based method of hand washing, and a higher proportion of headaches, sore throat, and loss of sense of taste, and the study further found that Neutrophil, creatinine, and alkaline phosphatase were significantly higher in COVID-19 patients. We authors appreciated the reviewer feedback given our manuscript.

 Major Comments

Line 282-283: Authors should run a logistic regression to test for association.

 Our study was titled with “Epidemiological, clinical and laboratory profile of patients presenting with severe acute respiratory syndrome (SARS-CoV-2) in Ethiopia: Comparative cross-sectional study”. It was not a study identifying the associated factors. And regarding the statistical analysis we have stated that, a descriptive analysis was done to see the characteristics of the study subjects. Continuous variables were dichotomized when clinically relevant. We estimated the mean and standard deviation for normally distributed continuous variables and used t-tests to see if there were any differences between the COVID-19 positive and COVID-19 negative patients. We used the chi-square test or Fisher's exact test to assess differences between groups for categorical and dichotomous data, in the data entry and analysis section of the manuscript. So, With all due respect to the reviewer, we believe that logistic regression to test for association was not important for the stated objective of the study. 

 Minor Comments

Abstract

Line 33: delete ‘was’ (the study aimed to compare). The comment is accepted. Based on the comment our revised manuscript is corrected. 

 Line 34: Add ‘of’ (clinical and laboratory profiles of patients). The comment is accepted. Based on the comment our revised manuscript is corrected.

 Line 38: Add ‘a’ (Using a structured questionnaire) The comment is accepted. Based on the comment our revised manuscript is corrected.

 Line 41: Replace (to see their laboratory profiles) with ‘for laboratory profile analysis’. The comment is accepted. Based on the comment our revised manuscript is corrected.

 Line 45: delete ‘were’ (participants tested positive) The comment is accepted. Based on the comment our revised manuscript is corrected.

 Methods

Line 111: what about those who are 18 years and below? Participants who were under 18 years were not our study subject. They were not included in our inclusion criteria. 

 Line 113: A suspected case is based on meeting one or more of either clinical or epidemiological link criteria. It is not clear why, according to the authors, patients were tested irrespective of whether they met the criteria or not. Sorry for the mistake, by now it is corrected as “Suspected patients were tested for SARS-CoV-2 if they met COVID-19 symptoms clinically or laboratory testing criteria of the National Ethiopian Public Health Guideline”, in the revised manuscript.

 Line 117: The sentence should be paraphrased. The comment is accepted. Based on the comment our revised manuscript is paraphrased.

 Line 134: Per the calculation, the minimum sample size must be 423. Thank you. Sorry for the mistake, by now it is corrected as a minimum sample size to be 423 and not 422 in the revised manuscript.

 Line 142: delete ‘an.’ The comment is accepted. Based on the comment our revised manuscript is corrected.

 Line 149: This statement contradicts that of line 113. The authors must clarify this. The comment is accepted. Based on the comment we clarified by no in our revised manuscript in the line 113.

 Line 156: Provide a reference for the protocol or the standard operating procedure.

 Thank you for your constructive comment. However, we have already cited our study protocol as a reference “24” in the main document and listed as “FMOH. National Comprehensive COVID-19 Management Handbook. Addis Ababa; 2020”, in the reference list.

 Line 182: With the biochemical and enzymatic tests, the blood was collected into which tube since an inappropriate tube can lead to wrong results. E.g., sodium fluoride (NaF) tubes and serum separation tubes (SST)

 Thank you again. The comment is taken and modification is done on the revised manuscript. For your information we have used serum separation tubes (SST).

 Line 192-193:= 215-216 Authors should state exactly what was done to prevent or minimize pre-analytical, analytical, and post-analytical errors. Specimen collection, handling, storage including temp, Assays used, quality control testing, and reliability of assays. The comment is accepted.

For laboratory analysis pre-analytical such as identifying the right patient and laboratory sample type and time, sample handling, patient and, material preparations, checking of quality control; in the analytical phase we follow starting samples have been logged into the lab. This phase is comprised of the lab diagnostic and testing procedures and instrument analysis. In the post-analytical stages, we monitored correctly the result dispatch, and verification of the results of the laboratory work was done to maximize the quality data that were proven in the Standard Operating Procedures (SOPs) of the laboratory.

 Results

Line 211: Indicate the figure. 

The comment is accepted. Based on the comment, we have corrected on the revised manuscript.

 Line 216: Authors should re-categorize the groups i.e., Occupation. I think some health professionals, as well as drivers, are Government employees.

-Education: The word illiterate is too offensive. Replace with no formal education.

College and above: replace with tertiary. 

The comment is accepted. Based on the comment, we have modified on the revised manuscript.

 Line 231: Authors must be consistent in reporting. Just use the percentages The comment is accepted. By now we have used percentages consistently in the revised manuscript.

 Line 260: Use a ‘differential blood count.’ The comment is accepted. We have modified the revised manuscript based on the comment as “Differential blood count”

 Discussion

Line 294-295: It is not clear which respiratory tract infections apart from Covid-19, that, the authors compare the clinical, laboratory, and other variables with. The comment is well taken. By now we have removed the phrase “Without a control group and limited COVID-19 test availability, we cannot ensure whether COVID-19 presents differently from other types of respiratory diseases” form the revised version of the manuscript.

 Line 296-297: the study site is a referral and treatment center. Most patients are likely to test positive for Covid-19. Moreover, some might be severely ill and have received some form of treatment before being referred to the facility. This can lead to bias in the study The comment is well taken. It is true that the hospital is a referral specialized hospital in Addis Ababa. Patients presenting with SARS-CoV-2 came to the center from various corners of the country outside of the capital city of Ethiopia. Additionally, the study included any acute respiratory illness (runny nose and sore throat) and at least one of the following symptoms: fever, cough, and shortness of breath, whereas previously known COVID-19 disease patients were excluded from the study, this concern incorporated as a limitation of the study.

 6. PLOS authors have the option to publish the peer review history of their article (what does this mean?). If published, this will include your full peer review and any attached files.

Do you want your identity to be public for this peer review? For information about this choice, including consent withdrawal, please see our Privacy Policy.

Reviewer #1: Yes: Comfort Dede Tetteh

Reviewer #2: No

We thank all editors and reviewers participated to review this manuscript. We authors have no any problem weather it was anonymously reviewed or not.

---

## [Decision Letter · Decision Letter 1]

2 Aug 2023

PONE-D-23-08723R1Epidemiological, clinical and laboratory profile of patients presenting with severe acute respiratory syndrome (SARS-CoV-2) in Ethiopia: Comparative cross-sectional studyPLOS ONE

Dear Dr. Gize,

Thank you for submitting your manuscript to PLOS ONE. After careful consideration, we feel that it has merit but does not fully meet PLOS ONE’s publication criteria as it currently stands. Therefore, we invite you to submit a revised version of the manuscript that addresses the points raised during the review process.

ACADEMIC EDITOR:

Title: Since you did not recruit participants without SARS-CoV-2, delete Comparative cross sectional study from the title

Line 41: specify the laboratory profiles you analyzed

Line 42: Revise …p. value less than 0.05 were to p-values less than 0.05… or p-values < 0.05

Line 44: SARS-CoV-2 and not SARS- CoV-2

Line 49: neutrophil and not Neutrophil

Line 50: Show mean and SD values instead of %. Also indicate the p-values so confirm significance

Line 54: do not capitalize Creatinine, Alkaline. Are trying to conclude that elevation of these markers confirms COVID-19? Please revise this statement

Lines 60/70: Define all abbreviations (COVID-19, CDC) on first mention

Lines 68/69: Try expressing these huge number in millions and billions for ease of appreciation

Lines 82-84: Were these parameters elevated or reduced?

Lines 131/132: Because we had no previous proportion of positivity during the study period, we took the positivity rate of RT-PCR COVID-19 to be 50%...refer readers to previous publication where this assumption was used

Lines 129-36: indicate the formula used to determine the sample size

Line 159: indicate the source of the Da’an Gene Corporation. Does the kit isolate NA or RNA? Specify

Lines 169/170: indicate the rtqPCR equipment used

Line 176: vacutainer

Lines 175/182: Indicate the volume of blood samples collected into each tube

Line 185: The instrument was fully automated and can do up to 60 assays like hemoglobin A1C (HBA1C), Alanine Aminotransferase (GPT), etc…statement not necessary

Line 191: indicate the Westgard rule used to assess the validity of the biochemistry values

Line 193/99: sentence too long. Revise

Line 211: include ethics approval reference number

Line 226: More than half of them were males.. add number (%)

Line 237: change figure 1 to (figure 1) same as line 241

Lines 243/44: what do you mean by higher, lower and normal values? Indicate actual mean values and p-values to show sig differences

Table 2: I am not sure the table is well presented. Which population does the normal values you indicated refer to? Of the 413 cases, 250 and 163 were positive and negative respectively. Compare the mean blood parameters between the positive and negative cases. In its current form, it doesn’t bring out a good outcome of the intended analysis.

Line 258: You use college in the test and tertiary school in table 3, please be consistent

Lines 258-60: patients with a positive COVID-19 test 259 were less likely to use the only alcohol-based method of hand washing (12.5% vs 87.5%, P<0.05),… how did you determine the less likelihood?

Table 3, under occupation, why is the frequency for government in parenthesis?

Table 3, why did you use fisher test for ‘Daily rubbing hands’ when all the frequencies were more than 5

Table 3: was Method of hand washing analyzed with chi or fisher exact?

Lines 265-68: better to write p=0.001 instead of P=0.001. Revise here and elsewhere

Table 5: T-test is more appropriate to analyze Table 5

Line 301: Obviously these changes will affect the discussion. Please revise the discussion after effecting these changes.

We look forward to receiving your revised manuscript.

Kind regards,

Enoch Aninagyei, PhD

Academic Editor

PLOS ONE

Journal Requirements:

Reviewers' comments:

Reviewer's Responses to Questions

**Comments to the Author**

1. If the authors have adequately addressed your comments raised in a previous round of review and you feel that this manuscript is now acceptable for publication, you may indicate that here to bypass the “Comments to the Author” section, enter your conflict of interest statement in the “Confidential to Editor” section, and submit your "Accept" recommendation.

Reviewer #1: All comments have been addressed

2. Is the manuscript technically sound, and do the data support the conclusions?

Reviewer #1: Yes

3. Has the statistical analysis been performed appropriately and rigorously? 

Reviewer #1: Yes

4. Have the authors made all data underlying the findings in their manuscript fully available?

Reviewer #1: Yes

5. Is the manuscript presented in an intelligible fashion and written in standard English?

Reviewer #1: Yes

6. Review Comments to the Author

Reviewer #1: Thank you for considering the comments suggested and addressing them accordingly to improve on the manuscript.

7. PLOS authors have the option to publish the peer review history of their article (what does this mean?). If published, this will include your full peer review and any attached files.

Reviewer #1: **Yes: **Comfort Dede Tetteh

---

## [Author Response · Author response to Decision Letter 1]

9 Nov 2023

View Letter

Date: Aug 02 2023 12:35PM

To: "Addisu Gize" konjoaddisu@gmail.com

From: "PLOS ONE" plosone@plos.org

Subject: PLOS ONE Decision: Revision required [PONE-D-23-08723R1]

PONE-D-23-08723R1

Epidemiological, clinical and laboratory profile of patients presenting with severe acute respiratory syndrome (SARS-CoV-2) in Ethiopia: Comparative cross-sectional study

PLOS ONE

Dear Dr. Gize,

Thank you for submitting your manuscript to PLOS ONE. After careful consideration, we feel that it has merit but does not fully meet PLOS ONE’s publication criteria as it currently stands. Therefore, we invite you to submit a revised version of the manuscript that addresses the points raised during the review process.

Response: We authors would like to thank for your invitation to submit our revised version of the manuscript, PONE-D-23-08723R1 

Epidemiological, clinical and laboratory profile of patients presenting with severe acute respiratory syndrome (SARS-CoV-2) in Ethiopia: Comparative cross-sectional study. In the revised version of the manuscript, we addressed points raised during the review process.

ACADEMIC EDITOR:

Title: Since you did not recruit participants without SARS-CoV-2, delete Comparative cross sectional study from the title

Response: The comment is accepted. By now we have removed “comparative cross sectional study from our revised manuscript title.

Line 41: specify the laboratory profiles you analyzed

Response: The comment is accepted. In the revised manuscript we have specified as “hematological tests like; white blood cell count, hematocrit, and platelet count; and biochemical and enzymatic tests like; aspartate transaminase (AST), creatinine, etc are mentioned/specified (line 42-44, within track change file of the abstract section revised manuscript).

Line 42: Revise …p. value less than 0.05 were to p-values less than 0.05… or p-values < 0.05

Response: Thank you! The comment is accepted. Based on the comment, we have corrected it (line 45, within track change file of the abstract section revised manuscript). 

Line 44: SARS-CoV-2 and not SARS- CoV-2

Response: The comment is accepted. Based on the comment, we have corrected it (line 47, within track change file of the abstract section revised manuscript).

Line 49: neutrophil and not Neutrophil

Response: The comment is accepted. Based on the comment, we have corrected it (line 52, within track change file of the abstract section revised manuscript).

Line 50: Show mean and SD values instead of %. Also indicate the p-values so confirm significance

Response: Thank you for the comment. Since each laboratory-related factor has a cut point where it may be classified as normal range, if it is in the normal, and if it is abnormal we categorized as abnormally high values (above the normal range), or low values (below the normal range), we analyzed the participant laboratory values as categorical variables(low, normal and high values). Examples include creatinine, alkaline phosphatase, and others. This categorical classification is clinically significant and applicable to treat the participants rather than comparing mean and SD laboratory values. 

line 54: do not capitalize Creatinine, Alkaline. Are trying to conclude that elevation of these markers confirms COVID-19? Please revise this statement

Response: Based on the comment we have revised it (line 58 within track change file of the abstract section revised manuscript).

Lines 60/70: Define all abbreviations (COVID-19, CDC) on first mention

Response: Thank you for your constructive comments. Based on the comment we have revised it (line 72 and 83 within track change file of the introduction section revised manuscript).

Lines 68/69: Try expressing these huge number in millions and billions for ease of appreciation

Response: Thank you for your constructive comments. Based on the comment we have revised it. (Lines 81 and 82 within track change file of the introduction section revised manuscript)

Lines 82-84: Were these parameters elevated or reduced?

Response: Thank you for your constructive comments. They are elevated during infection (Line 96 within track change file of the introduction section revised manuscript).

Lines 131/132: Because we had no previous proportion of positivity during the study period, we took the positivity rate of RT-PCR COVID-19 to be 50%...refer readers to previous publication where this assumption was used

Response: With all due respect to the editors’, we believe that this point is not correct; we had no previous study means, it is recommended by the statistics science to use the formula (50% proportion) to calculate the sample size determination for the current study. 

Lines 129-36: indicate the formula used to determine the sample size

Response: The formula is indicated below (written on lines 152-158 within track change file of the method section revised manuscript). 

n = (zα/2)2 p (1-p) (1.96)2 0.5(1-0.5) = 384 study subjects.

 d2 (0.05)2 

Where: n = minimum sample size required, 

 P = estimated proportion of patients with SARS-CoV-2 in the study population, and taking 10% non-response rate, the final sample size become 423 participants. 

d= the margin of sample error, zα/2= the standard normal variable at 1-α/2 confidence level and we used consecutive sampling technique was used to select the study population.

Line 159: indicate the source of the Da’an Gene Corporation. Does the kit isolate NA or RNA? Specify

Response: It is sourced from Guangdong, China, and can isolate RNA (Line 185 within track change file of the method section revised manuscript).

Lines 169/170: indicate the rtqPCR equipment used

Response: A quantitative PCR instrument is a machine that amplifies and detects DNA or RNA. In the case of SARS-CoV-2, the rtqPCR combines the functions of a thermal cycler and a fluorimeter, enabling the process of quantitative PCR (Lines 198-200 of within track change file of the method section revised manuscript).

Line 176: vacutainer

Response: The comment is accepted. Based on the comment we have corrected (written on line 206 within track change file of the method section revised manuscript).

Lines 175/182: Indicate the volume of blood samples collected into each tube

Response: The comment is accepted. Based on the comment we have corrected the volume of blood taken from each study subject as 5ml (written on line 205 within track change file of the method section revised manuscript).

Line 185: The instrument was fully automated and can do up to 60 assays like hemoglobin A1C (HBA1C), Alanine Aminotransferase (GPT), etc…statement not necessary

Response: Thank you for your constructive comment. Now we have removed the unnecessary statement from the revised manuscript.

Line 191: indicate the Westgard rule used to assess the validity of the biochemistry values

Response: We used Westgard rules to assess the validity of the biochemistry values and the rule recommends tolerating only 12s i.e., one measurement exceeds 2 standard deviations either above or below the mean of the reference range (written on line 230-232 within track change file of the method section revised manuscript).

Line 193/99: sentence too long. Revise

Response: Thank you for your constructive comment. Now we have revised it. 

Line 211: include ethics approval reference number

Response: Thank you. The comment is taken and by now we have added ethics approval reference number to the revised manuscript (written on line 245 within track change file of the method section revised manuscript).

Line 226: More than half of them were males.. add number (%) 

Response: The comment is taken and edited as; n=244(59.1%) (Written on line 279 within track change file of the result section revised manuscript)

Line 237: change figure 1 to (figure 1) same as line 241

Response: The comment is accepted and edited as (figure 1) and (figure 2) in lines 289 and 293 within track change file of the method section revised manuscript

Lines 243/44: what do you mean by higher, lower and normal values? Indicate actual mean values and p-values to show sig differences

Response: With all due respect to the editor, this point is not correct. 

Every laboratory test has its own reference range. For example the normal reference ranges for hemoglobin count for adult population described in different book as “13.6-17.5 g/dL” of blood. This means, below the normal limit (˂13.6g/dL) i.e., we will call it lower or abnormally low result, clinically leads to anemia, and also if it is above the normal limit .i.e., greater than 17.5g/dL, also indicate abnormally high hemoglobin level of the patient. So, we have to classify our study participants based on these reference ranges as how many of the participants are in the normal reference ranges. i.e., 13.6-17.5g/dL for hemoglobin laboratory results as example given above, and other laboratory profiles too based on their normal reference ranges at the time of their presentation for diagnosis. Regarding to the p-value and significance, it is listed in the table 5 for each laboratory parameter.

Table 2: I am not sure the table is well presented. Which population does the normal values you indicated refer to? Of the 413 cases, 250 and 163 were positive and negative respectively. Compare the mean blood parameters between the positive and negative cases. In its current form, it doesn’t bring out a good outcome of the intended analysis.

Response: This table 2 is the laboratory test results of all patients at the time of presentation. And categorized as how many of the participants were in the normal reference range on each laboratory test results, how many them were below the reference range (low) and how many of them were elevated (high) their result from each reference range. For example from this table in the case of white blood cell count (n=319 participants) at the time of presentation, of which 14(4.4%) of them were below the normal reference range level, lower than lower limit (lower than the LL i.e., less than 3.4-10 x109/L, we will call it leukopenia, and 119(37.3%) had greater than 10 x109/L white blood cell count, we will call it leukocytosis or abnormally elevated number of white blood cells whereas the majority category of the participants, 186(58.3%) had normal values (fall in the normal reference interval), ( table 2).

Line 258: You use college in the test and tertiary school in table 3, please be consistent

Response: The comment is accepted. By now we have consistent having tertiary school (in the line 317 within track change file of the result section revised manuscript

Lines 258-60: patients with a positive COVID-19 test 259 were less likely to use the only alcohol-based method of hand washing (12.5% vs 87.5%, P<0.05),… how did you determine the less likelihood?

Response: Thank you for the valuable comment. Since we conducted the analysis using chi-square test, we can’t use the word less likely and we have corrected this in the manuscript using the chi squire test interpretation (line 317-321). 

Table 3, under occupation, why is the frequency for government in parenthesis?

Response: Sorry for the mistake. By now we have removed it.

Table 3, why did you use fisher test for ‘Daily rubbing hands’ when all the frequencies were more than 5

Response: Thank you so much for your valuable comment. We have re-run the analysis again and the test fits for chi-square and not for fisher exact test. It is corrected accordingly. 

Table 3: was Method of hand washing analyzed with chi or fisher exact?

Response: Thank you for your valuable comment. We took the chi squire test p-value; however, it must be fisher exact test. We have corrected accordingly. 

Lines 265-68: better to write p=0.001 instead of P=0.001. Revise here and elsewhere

Response: Thank you for your constructive comment. Now we have corrected.

Table 5: T-test is more appropriate to analyze Table 5

Response: Thank you for your comment. Clinically it is very important to classify laboratory results or values as low, normal and high. Hence, we have considered the laboratory profile as a categorical covariate variable (categorized as below the normal reference interval, normal reference interval and above reference interval). Also the data was analyzed using the chi squire and fisher exact test statistics. 

Line 301: Obviously these changes will affect the discussion. Please revise the discussion after effecting these changes.

Response: Yes, thank for your concern and constructive comments. By now we have gone through in all section of the manuscript (discussion, conclusion and recommendation). 

Response: We included a point-by-point response within the 'Response to Reviewers' box in the submission system and modifications are highlighted in the track changes of the original manuscript, and we uploaded also the clean or unmarked version of our revised paper without tracked changes, file labeled as ‘Manuscript’.

Response: No need of change about our manuscript regarding to financial disclosure

We look forward to receiving your revised manuscript.

Kind regards,

Enoch Aninagyei, PhD

Academic Editor

PLOS ONE

Response: Thank you Enoch Aninagyei, PhD, Academic Editor. We have revised based on the raised points. 

Reviewers' comments:

Reviewer's Responses to Questions

Comments to the Author

1. If the authors have adequately addressed your comments raised in a previous round of review and you feel that this manuscript is now acceptable for publication, you may indicate that here to bypass the “Comments to the Author” section, enter your conflict of interest statement in the “Confidential to Editor” section, and submit your "Accept" recommendation.

Reviewer #1: All comments have been addressed

Response: Thank you for your scientific contribution to improve our manuscript.

2. Is the manuscript technically sound, and do the data support the conclusions?

Reviewer #1: Yes

Response: Thank you.

3. Has the statistical analysis been performed appropriately and rigorously? 

Reviewer #1: Yes

Response: Thank you.

4. Have the authors made all data underlying the findings in their manuscript fully available?

Reviewer #1: Yes

Response: Thank you.

5. Is the manuscript presented in an intelligible fashion and written in standard English?

Reviewer #1: Yes

Response: Thank you.

6. Review Comments to the Author

Reviewer #1: Thank you for considering the comments suggested and addressing them accordingly to improve on the manuscript.

Response: Thank you.

7. PLOS authors have the option to publish the peer review history of their article (what does this mean?). If published, this will include your full peer review and any attached files.

Do you want your identity to be public for this peer review? For information about this choice, including consent withdrawal, please see our Privacy Policy.

Reviewer #1: Yes: Comfort Dede Tetteh

Response: Dear Comfort Dede Tetteh, Thank you so much.

---

## [Editor Report · Decision Letter 2]

17 Nov 2023

Epidemiological, clinical and laboratory profile of patients presenting with severe acute respiratory syndrome (SARS-CoV-2) in Ethiopia

PONE-D-23-08723R2

Dear Dr. Gize,

We’re pleased to inform you that your manuscript has been judged scientifically suitable for publication and will be formally accepted for publication once it meets all outstanding technical requirements.

Kind regards,

Enoch Aninagyei, PhD

Academic Editor

PLOS ONE
---

## [Editor Report · Acceptance letter]

23 Nov 2023

PONE-D-23-08723R2 

Epidemiological, clinical and laboratory profile of patients presenting with severe acute respiratory syndrome (SARS-CoV-2) in Ethiopia 

Dear Dr. Gize:

I'm pleased to inform you that your manuscript has been deemed suitable for publication in PLOS ONE. Congratulations! Your manuscript is now with our production department. 

Kind regards, 

on behalf of

Dr Enoch Aninagyei 

Academic Editor

PLOS ONE